# Ureterocystoplasty in Boys with Valve Bladder Syndrome—Is This Method Still up to Date?

**DOI:** 10.3390/children10040692

**Published:** 2023-04-06

**Authors:** Aybike Hofmann, Alexandros Ioannou, Pirmin Irenaeus Zöhrer, Wolfgang H. Rösch

**Affiliations:** 1Department of Pediatric Urology in Cooperation with University Medical Center Regensburg, Hospital Barmherzige Brüder, Clinic St. Hedwig, 93049 Regensburg, Germany; 2Department of Pediatric Surgery, Hospital Barmherzige Brüder, Clinic St. Hedwig, 93049 Regensburg, Germany

**Keywords:** posterior urethral valves, urinary diversion, bladder augmentation, congenital lower urinary tract obstruction, kidney function, long-term outcome

## Abstract

Boys with valve bladder syndrome (PUV) require adequate treatment of the lower urinary tract to preserve renal function and improve long-term outcomes. In some patients, further surgery may be necessary to improve bladder capacity and function. Ureterocytoplasty (UCP) is usually carried out with a small segment of intestine or, alternatively, with a dilated ureter. Our aim was to evaluate the long-term outcomes after UCP in boys with PUV. UCP had been performed in 10 boys with PUV at our hospital (2004–2019). Pre- and postoperative data were evaluated in relation to kidney and bladder function, the SWRD score, additional surgery, complications, and long-term follow-up. The mean time between primary valve ablation and UCP was 3.5 years (SD ± 2.0). The median follow-up time was 64.5 months (IQR 36.0–97.25). The mean increase in age-adjusted bladder capacity was 25% (from 77% (SD ± 0.28) to 102% (SD ± 0.46)). Eight boys micturated spontaneously. Ultrasounds showed no severe hydronephrosis (grade 3–4). The SWRD score showed a median decrease from 4.5 (range 2–7) to 3.0 (range 1–5). No conversion of augmentation was required. UCP is a safe and effective approach to improve bladder capacity in boys with PUV. In addition, the possibility of micturating naturally is still maintained.

## 1. Introduction

In boys with valve bladder syndrome (PUV), the main treatment aim is to relieve the pressure on and the obstruction of the urinary tract to improve both bladder and kidney functions. The initial treatment of choice includes primary valve ablation followed by more conservative management or temporary urinary diversion. If these treatments fail, surgical bladder augmentation has to be considered to prevent further deterioration of the urinary tract [1].

Traditionally, bladder reconstruction is carried out by using a reconfigured segment of the intestine. Despite the effectiveness of this method in increasing bladder volume, it is not without side effects [2]. The interaction of urine with functioning ileal or sigmoid mucosa may cause metabolic disturbances, and intestinal mucus secretion may lead to stone formation. Further adverse events are spontaneous perforation as well as an increased risk of malignancy. Additionally, boys with PUV are often unable to receive intestinal augmentation because of renal insufficiency. 

As a consequence, a dilated ureter is usually used for bladder reconstruction in boys with PUV [1,2]. This procedure was first described by Eckstein and Martin in 1973 [3]; however, it took twenty years until further reports were published, for example, by Bellinger and Churchill et al. [4,5]. These publications were followed by several reports with promising results at the beginning of the new millennium. Since then, however, UCP seems to have gone out of fashion. 

Therefore, we evaluated the efficacy and safety of UCP as well as health-related quality of life for long-term outcomes in boys with PUV who had undergone UCP at our hospital.

## 2. Materials and Methods

The study was approved by the institutional ethics committee (no. 22-2778-101), and informed consent was obtained from the parents or legal representatives of each prospectively examined participant. 

We identified all children with PUV who had received UCP at our department (full member of ERN-eUROGEN) between 2004 and 2019. All patients had previously undergone successful primary valve ablation. Medical records were retrospectively reviewed in terms of the indication and timing of UCP, preoperative kidney function, history of kidney transplantation, preoperative bladder cystometry, ultrasound of the urinary tract, and voiding cystourethrogram (VCUG). Additionally, all patients were invited to take part in a follow-up examination after a median of 83 months (IQR 37.00–99.75) as part of the study. Two patients who had subsequently moved abroad were not able to participate in the follow-up examination. Instead, we included the last follow-up examination at our clinic 3 and 4 years after surgery. The follow-up examination included blood analysis (serum creatinine and cystatin c), ultrasounds of the urinary tract, uroflowmetry or invasive urodynamic measurement, and VCUG as appropriate. Exclusion criteria were missing informed consent, a follow-up period of less than three years, and coexisting syndromes. 

Additionally, the German version of the Kinder Lebensqualität Fragebogen (KINDL^®^) questionnaires (kiddy: 4–6 years; kid: 7–13 years; kiddo: 14–17 years) for children and the proxy version for parents were implemented as a QoL measure. This self-reported form consists of 24 items equally distributed into six subscales: physical well-being (e.g., felt tired and worn-out), emotional well-being (e.g., felt alone, was scared), self-esteem (e.g., was proud of myself), well-being related to family (e.g., got on well with my parents), well-being related to friends/peers (e.g., other kids liked me), well-being related to school (e.g., was afraid of receiving bad marks), and a disease perception subscale for chronic diseases (e.g., being afraid that my disease might get worse; being sad because of my disease; being unable to cope well with my disease; being afraid of others noticing my disease; and being afraid of missing something at school because of my disease). Each item is rated on a 5-point scale (1 = never, 2 = seldom, 3 = sometimes, 4 = often, and 5 = always) and addresses experiences over the previous week. Mean scores are calculated for each of the six subscales and for the total scale. All subscales were then linearly transformed to a 0–100 scale. Higher scores represent a better QoL [6]. Details about the questions can be found on the KINDL web site (www.kindl.org (accessed on 22 March 2023). 

### 2.1. Surgical Technique

Because all patients had a valve unilateral reflux and dysplasia (VURD) syndrome, the initial treatment consisted of laparoscopic nephrectomy of the non-functioning kidney, and the entire ureter was used for UCP. Removal of the kidney was carried out using the extraperitoneal approach. The incision of choice for UCP is the Pfannenstiel method, except for patients who are expected to undergo a kidney transplant after UCP. In such patients, incision was carried out using the unilateral pararectal hockey-stick method to avoid a second scar. The dilated ureter (>10 mm) was dissected from the surrounding tissues with care to preserve its blood supply. The ureter was then incised at the anterior side and sutured together in a U-shape forming a sheet (Figure 1). The bladder was then incised starting from the vesicoureteral junction towards the posterior wall, and the ureter sheet was sutured to the bladder wall.

If necessary, the patient simultaneously received a continent (Mitrofanoff-stoma) or incontinent (vesicostomy) stoma as well as an antireflux plasty (ARP) of the contralateral ureter. Two patients had received a kidney transplant prior to UCP, while three patients were augmented in preparation for a transplant. Detailed information is shown in Figure 2.

### 2.2. Statistical Analysis

Continuous variables are shown as median (interquartile range), median (range: minimum–maximum), or mean (±standard deviation) as appropriate. Categorical variables are expressed as counts and percentages. Comparisons of non-normally distributed continuous paired variables amongst the groups were performed with the Wilcoxon signed rank test and comparisons of categorical variables between groups with the Pearson’s x^2^ test or the Mann–Whitney U test as appropriate. The KINDL questionnaire was analyzed using the tool provided by the inventor. Additionally, a one sample t-test was used for statistical evaluation, as only mean values but no raw data were available for the control group. All analyses were conducted using SPSS^®^, version 28.0 (IBM Corp., Armonk, NY, USA).

## 3. Results

At our department, 11 patients had received UCP between 2004 and 2019; 1 patient was excluded from analysis because of a coexisting syndrome that co-affected the bladder. The indications for UCP were a combined low capacity and low compliance bladder with contralateral VUR in eight patients and without contralateral VUR in two patients. Median age at UCP was 3.8 years (IQR 2.2–6.1). None of the patients developed any immediate postoperative complication.

All patients were followed-up after a median period of 64.5 months (IQR 36.0–97.25). Initial VUR on the contralateral kidney was present in eight patients. Two patients showed persisting VUR after surgery: one patient in the subsequently transplanted kidney, the other in the contralateral dysplastic kidney, which additionally caused recurrent urinary tract infection. Overall, the median decrease in the SWRD score was 1.5 (from 4.5 (range 2–7) to 3.0 (range 1–5)) (*p* = 0.147) (Figure 3). The mean age-adjusted bladder capacity increased from 77% (±0.28) before UCP to 102% (±0.46) (*p* = 0.139) after UCP, resulting in a 1.3-fold enlargement of the bladder.

After surgery, renal ultrasounds did not show severe hydronephrosis (grade 3–4) in any of the patients; seven (70%) patients showed mild hydronephrosis (grade 1–2), whereas three (30%) patients did not. Median parenchymal thickness of the contralateral kidneys of boys who did not undergo a transplant was 17.4 mm (±2.8). 

With regard to kidney function, a mean increase in the glomerular filtration rate (GFR) from 52.2 mL/min/1.73 m^2^ (range 42–63) to 57.7 mL/min/1.73 m^2^ (range 50–65.3) and a median increase from 8 mL/min/1.73 m^2^ (range 6–10) to 57.9 mL/min/1.73 m^2^ (range 45–70.7) was found in boys who had preoperatively undergone a transplant. A median decrease from 95 mL/min/1.73 m^2^ (range 20–159) to 67.8 mL/min/1.73 m^2^ (range 22.1–101.6) was seen in boys without a transplant.

With regard to the stage of chronic kidney disease, the two boys with a preoperative transplant remained stable (CKD 3), and the boys with a postoperative transplant showed an improvement in the CKD stage from CKD 5 to CKD 2/3. Two boys without a transplant were lost to long-term follow-up; of the other four boys, two showed stable values (CKD 1 and 4) and 2 a decline from CKD 1/2 to CKD 2/3. 

Free flow was present in eight (80%) boys; none of these boys needed any further urodynamic measurements. Invasive urodynamic measurement was conducted in five (50%) boys, in four (80%) boys via the existing stoma. Four (80%) patients showed normal bladder compliance and 1 (20%) patient persisting low compliance. No bladder stones or recurrent urinary tract infections were observed. None of the patients required a conversion of the augmentation. Detailed data on the current nephrological status and bladder outcome are shown separately for each patient in Table 1.

Additionally, KINDL questionnaires were answered by seven patients and their parents. Out of these seven children, four answered the “kid-version”, one answered the “kiddo-version”, and the two youngest children answered the “kiddy-version”. All parents answered the same proxy version of the questionnaire. For the control group, we used older data from a collective of Hamburg schoolchildren. For the present study, our data were selectively compared only with the data of boys of the control group, namely 762 boys. The questionnaires for the youngest group were adapted to the cognitive level of that age group and only included statements on the overall quality of life and disease-specific quality of life. Additionally evaluated items for the remaining patients in our cohort and the control group are shown in Table 2. 

In summary, the data show that both the two younger boys and the five older boys reported at least equal or even better quality of life for nearly all items. In terms of physical well-being and school-related quality of life, self-assessments were significantly better. A somewhat lower quality of life was reported only in relation to their own family. On average, parents rated the quality of their children’s lives lower than the children themselves; the only exception was the disease-specific sub-scale. The greatest difference in the ratings between parents and children was in mental and physical well-being and in quality of life in connection with friends.

## 4. Discussion

In 1870, Tolmatschew had already recognized that PUVs are not an isolated anatomical defect [7] but a complex pathological entity that has been termed ‘valve bladder syndrome’ (PUV) by Mitchel [8]. This term emphasizes the importance of achieving optimum interaction between bladder function and the upper urinary tract to preserve renal function in boys with PUV.

There is general agreement that primary valve ablation has become the initial treatment of choice [9,10]. Despite the success of early valve ablation to release the anatomical obstruction, the percentage of morbidities is still significant, for example, persistent bladder dysfunction in 70–90% of patients or end-stage renal failure [11]. Bladder augmentation has to be especially considered in patients who do not respond to either of the conservative and surgical treatment options (e.g., vesicostomy) to achieve adequate bladder function. 

Despite the well-known multiple side effects (metabolic acidosis, intestinal mucus secretion, stone formation, abnormalities of calcium metabolism, and malignancy due to bringing urine in contact with functioning ileal or colonic mucosa), intestinal segments are still the most frequently used tissue for bladder augmentation [2]. Yet, a ureter may still be the tissue of choice for a variety of reasons, particularly in boys with PUV.

Several studies have already reported successful bladder augmentation with a ureter. The various techniques available include the use of the entire ureter and removal of the ipsilateral kidney, the use of only the distal ureter on one side and reimplanting the proximal ureter in the bladder, transureteroureterostomy, and the use of both distal ureters [2,12,13]. When comparing the results of UCP and ileocystoplasty (eight children each in two groups), Landau et al. found a small difference in the degree of increased BC between the groups, favoring UCP [14]. In contrast, Podesta conducted a similar study and found ileocystoplasty to produce a higher degree of augmentation than UCP [15]. Tekgül et al. used various techniques for ureteroplasty in six children (two with PUV) and on average achieved an increase in bladder capacity of 263% [13]. Similar results were described by Youssif et al., who found a significant increase in BC (mean preop 107 mL, postop 288 mL) in eight boys with PUV after UCP [1]. The alternatives to enterocystoplasty were critically reviewed by Gonzalez et al., who reported a 1.83–3.75-fold degree of augmentation after UCP [2]. In our study, a 1.3-fold increase in BC was achieved by using the entire dilated ureter and by performing ipsilateral nephrectomy. 

One reason for the comparatively lower increase in BC in our study may be that our follow-up examination was mainly performed in prepubertal boys (mean 9.1 years), i.e., before the characteristic conversion of PUV bladder dysfunction from high-pressure low-compliant bladders and overactive bladders to myogenic-failure high-capacity bladders [16,17].

In addition to the increase in BC, most authors have reported improved bladder compliance after UCP. Tekgül et al. stated a 360% improvement in compliance. Similar results were demonstrated by Johal et al., who measured a mean increase in BC from 2.1 mL/min H_2_O to 16.2 mL/cm H_2_O in 17 children (10 PUV, 3 BE, 2 NVD). Youssif et al. also found significantly improved bladder compliance; however, none of the patients had achieved compliance >20 cm H_2_O [1]. In our cohort, normal bladder compliance was found in 4/5 (80%) boys. All boys in whom urodynamic measurement had been renounced showed an unsuspicious free flow pattern.

Hence, none of our patients required conversion to enterocystoplasty. Only one patient with persisting low-compliance bladder (<4 cm H_2_O) is planned to undergo a second ureteroplasty with the contralateral megaureter. Similar results were observed by Youssif et al. and Tekgül et al., who also saw no need for converting to enterocystoplasty [1,13]. Johal et al. reported a conversion rate of 24% in his cohort, assuming worse outcomes in patients with a neurologically damaged bladder than in patients with PUV [12]. 

Another matter of debate has been the timing of renal transplantation in relation to bladder augmentation. In 2013, López Pereira et al. demonstrated that bladder augmentation does not negatively affect renal transplants [18]. Most of their patients (10/12) were augmented with a ureter. Amesty et al. recently conducted a review to determine prognostic factors related to managing bladder dysfunction in boys with PUV and a renal transplant but did not find any clear indication regarding the time of bladder augmentation in relation to the time of transplantation [19]. These findings are consistent with our results. None of our patients experienced a graft loss or severe deterioration in graft function regardless of the time of UCP. In consideration of the long-term outcomes, we did not observe any difference in the development of CKD between the pre-or post-UCP transplanted boys.

Although PUV caused a decrease in GFR in the not-transplanted group, the majority of our patients showed an improved or stable GFR and CKD status.

Health-related quality of life (HQoL) is still an underestimated topic in boys with PUV. This lack of consideration may be related to the fact that the reduction in the high mortality rate was only made possible by the improvements in prenatal medicine over the past few years. Nowadays, boys with PUV are able to reach an age in which HQoL is of growing importance. In the past decade, the scientific group around Taskinen published some studies regarding HQoL, especially in adults with PUV [20,21]. The researchers found that the incidence of lower urinary tract symptoms (LUTS) is two-fold higher in adult patients with PUV. Accordingly, general quality of life seems only to be impaired in case of urinary incontinence or renal insufficiency [20].

To achieve better objectivity, we decided to use the validated KINDL^®^ questionnaire instead of a self-designed questionnaire. Yet, the evaluation of our patients is challenging because of the lack of specific questionnaires for boys with PUV or bladder augmentation. Unfortunately, the incidence of LUTS was not measured. Despite these limitations, it is remarkable that the patients who had undergone various operations because of their underlying disease rated their quality of life as being similar to that of healthy peers. The same results were observed by Jalkanen et al. in their study of adult patients with PUV who also considered their overall quality of life similar to that of the general population.

Patients in our collective subjectively rated their physical well-being as good, while, in comparison, their parents expressed a more pessimistic opinion in almost all items. This discrepancy may be due to the parents’ increased concern for their children in the face of an illness with far-reaching and lifelong consequences such as kidney failure.

The limitations of the study are its partially retrospective design and the low number of patients analyzed; the correlation with a high number of subjects in the reference group of the KINDL questionnaire in particular only allows limited conclusions. Yet, to our knowledge, this study is one of the few studies on the long-term follow-up of UCP in boys with PUV and has a representative number of patients considering the rare occurrence of this disease. Nevertheless, a prospective study with larger sample sizes should be performed to reevaluate our results.

## 5. Conclusions

UCP remains a safe and reliable approach that is highly accepted by patients and parents, especially in the case of boys with PUV who often show a megaureter and a non-functioning kidney.

In addition to the improvement in bladder function and the capacity to prevent severe deterioration of the upper urinary tract, the possibility to micturate naturally is still preserved in most patients. Nevertheless, UCP should be reserved for selected cases.

## Figures and Tables

**Figure 1 children-10-00692-f001:**
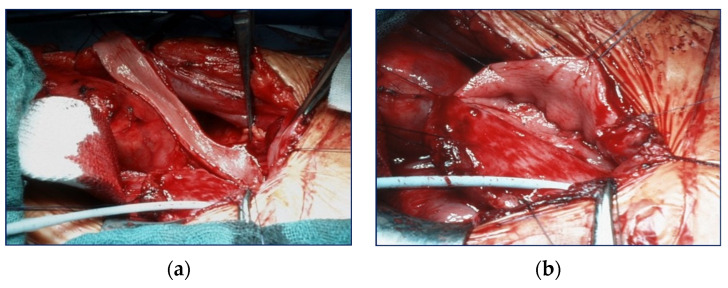
Anteriorly incised megaureter (**a**); U-shaped megaureter (**b**).

**Figure 2 children-10-00692-f002:**
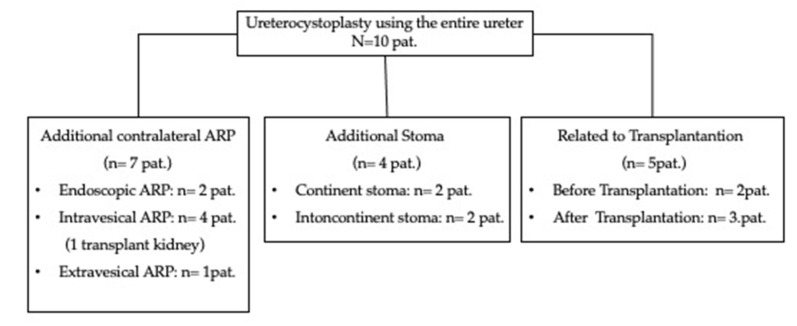
Distribution of the performed procedures.

**Figure 3 children-10-00692-f003:**
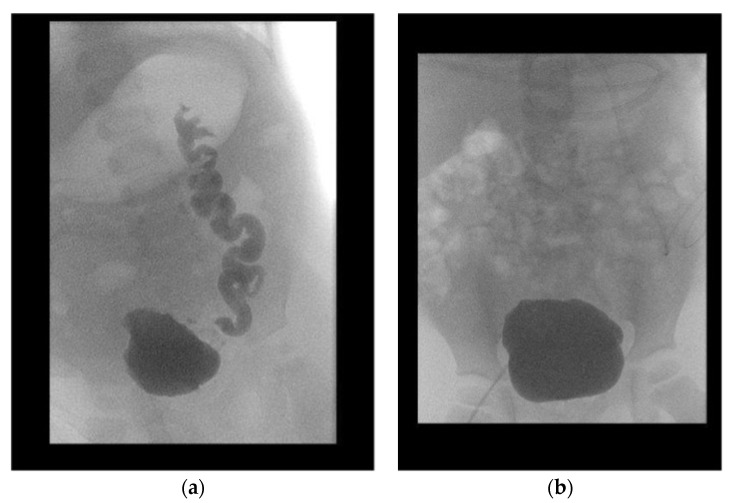
Pre- and postoperative voiding cystourethrograms of one patient (**a**,**b**). Preoperatively, severely trabeculated bladder and vesicoureteral reflux (**a**). Normalization of bladder configuration and decrease in vesicoureteral reflux at 1-year follow-up (**b**).

**Table 1 children-10-00692-t001:** Patient demographics and results of ureterocystoplasty of each patient.

Pat. No	Age Yrs.	Kidney					Bladder								
		Trans- Plant Status	GFR Preop	CKD Preop	GFR Postop	CKD Postop	BC Preop	BC Preop % Age-Related	BC Postop	BC % Postop Age-Related	Post Void Residual	Compliance Postop	Outcome	Flow Curve Shape	Medication
1	12	preop trans- plant	42	3	50	3	100	0.83	202	0.67	10	N/A	no CIC	staccato-shaped	no med.
2	10	no trans- plant	97	1	n		130	0.62	320	0.97		N/A	lost in long term follow-up		no med.
3	6	no trans-plant	89	2	n		125	1.04	240	1.14		N/A	lost in long term follow-up		no med.
4	9	preop trans-plant	63	2	65.31	2	200	0.95	329	1.4	85	15	on CIC via Mitrofanoff stoma	bell-shaped	Oxybutynin
5	4	no trans-plant	20	4	22.1	4	60	1	90	0.6	N/A	>20	no CIC, vesicostomy	N/A	no med.
6	13	postop trans-plant	10	5	45	3	35	0.23	180	0.98	0	>20	on CIC	N/A	Mictonetten
7	10	no trans-plant	65	2	53	3	90	1	217	0.66	24	N/A	no CIC	plateau-shaped	no med.
8	12	no trans-plant	125	1	82.5	2	100	0.56	230	1.52	0	>20	no CIC	bell-shaped	no med.
9	6	postop trans-plant	6	5	70.7	2	25	0.42	110	0.52	N/A	4	Left-sided nephrectomy with 2nd UCP and Mitrofanoff stoma	N/A	no med.
10	9	no trans-plant	159	1	101.6	1	60	1	559	1.86	12	N/A	no CIC	staccato-shaped	no med.

**Table 2 children-10-00692-t002:** Life quality items compared between the control group of German pupils and children with PUV + UCP and their parents.

	Total Quality (Mean)	Physical Well-Being (Mean)	Mental Well-Being (Mean)	Self-Esteem (Mean)	Family (Mean)	Friends (Mean)	School (Mean)	Disease (Mean)
Control	75.46	76.87	81.57	65.26	82.02	78.30	68.95	64.46
Augmented Children	79.31	90	85	71.25	78.75	88.75	82.5	79.17
*p*-value	0.350	**0.036**	0.464	0.441	0.638	0.142	**0.021**	0.157
Parents	76.94	81.25	78.57	68.75	75	79.46	78.57	86.31

Significance (*p* < 0.05) is highlighted in bold.

## Data Availability

The data used to support the findings off this study are available from the corresponding author upon request.

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
