# Peer review of "Ureterocystoplasty in Boys with Valve Bladder Syndrome—Is This Method Still up to Date?"

_children, 2023, doi:10.3390/children10040692_

Round 1
Reviewer 1 Report
Authors should be congratulated for the topic addressed. Valve bladder syndrome is a challenging-to-treat disease, worthy of interest by physicians to ensure the optimal functional outcome and to protect renal function from acute and chronic injuries. The manuscript is well-written and highlights important considerations on ureterocystoplasty. However several noteworthy considerations must be considered:
- Are data available on eGF, and creatinine, after a longer period after treatment?
- During the follow-up was the necessity of ureteral stent occur to treat the hydronephrosis?
- What is the rate of UTi after treatment?
- Are data available on the urodynamics studies performed to evaluate neo-bladder functionality?
- Authors should read this novel paper on the microbiome of Della Corte et al. It highlights several interesting points of view on the variety of bladder microenvironments (PMID: 36867096)
A major revision is required.
Author Response
Dear Reviewer 1,
we highly appreciate the chance to further work on our manuscript in order to have it published in MDPI Children.
Your comments proved to be helpful and constructive, and we feel able to respond accordingly to each of them. Please find a point-by-point response, including information on the changes made to the manuscript below.
Comments are in italic, changes to the manuscript have been implemented by using the ´Track Change` function.
We would like to express our gratitude for your fair and thoughtful review, raising important points yet underestimated in our manuscript.
Authors should be congratulated for the topic addressed. Valve bladder syndrome is a challenging-to-treat disease, worthy of interest by physicians to ensure the optimal functional outcome and to protect renal function from acute and chronic injuries. The manuscript is well-written and highlights important considerations on ureterocystoplasty. However several noteworthy considerations must be considered:
- Are data available on eGF, and creatinine, after a longer period after treatment?
eGF Data of each patient are already shown in table 1. eGF was calculated either by creatinine or by cystatin c. The postoperative measurement was performed after a median period of 64.5 months.
The following paragraph was additionally added to the manuscript:
With regard to kidney function, a mean increase in the glomerular filtration rate (GFR) from 52.2 ml/min/1.73m2 (range 42-63) to 57.7 ml/min/1.73m2 (range 50-65.3) and a median increase from 8 ml/min/1.73m2 (range 6-10) to 57.9 ml/min/1.73m2 (range 45-70.7) was found in boys who had preoperatively undergone a transplant. A median decrease from 95 ml/min/1.73m2 (range 20-159) to 67.8 ml/min/1.73m2 (range 22.1-101.6) was seen in boys without a transplant.
- During the follow-up was the necessity of ureteral stent occur to treat the hydronephrosis?
As we only had patients with mild hydronephrosis during the follow-up none of the patients required a ureteral stent placement.
- What is the rate of UTi after treatment?
None of the patients had a UTI after treatment. This result was added to the manuscript.
- Are data available on the urodynamics studies performed to evaluate neo-bladder functionality?
Data of urodynamic studies were available in. 5 patients. The results are already mentioned in the manuscript as follows:
Invasive urodynamic measurement was conducted in 5 (50%) boys, in 4 boys via the existing stoma. 4 (80%) patients showed a normal compliance of the bladder, persisting low compliance was seen in 1 (20%) patient.
- Authors should read this novel paper on the microbiome of Della Corte et al. It highlights several interesting points of view on the variety of bladder microenvironments (PMID: 36867096)
Thank you very much for this recommendation. We received the manuscript from the author and I read it with great interest.
We totally agree that there is a different microbiome in patients augmented with ileum. This is one reason why we prefer the usage of the ureter (if possible), because you have much less infection due to using the same tissue.
Reviewer 2 Report
Thank you for submitting your work to our Journal.
I suggest you do another English proofreading of your paper.
The main limit I see is the very small series, which you further divided according to the age group, leading to even smaller numbers in each group. Then you compared your results with a series of 762 patients. For most aspects, statistical significance was not obtained. I suggest you include some comments on these aspects.
Since your last patients was operated less than four years ago, I see this as medium term follow up.
Your conclusion is too generic, you should draw conclusions based on your study. Maybe answer at your question in the title of the paper.
Author Response
Dear Reviewer 2,
we highly appreciate the chance to further work on our manuscript in order to have it published in MDPI Children.
Your comments proved to be helpful and constructive, and we feel able to respond accordingly to each of them. Please find a point-by-point response, including information on the changes made to the manuscript below.
Comments are in italic, changes to the manuscript have been implemented by using the ´Track Change`function.
We would like to express our gratitude for your fair and thoughtful review, raising important points yet underestimated in our manuscript.
I suggest you do another English proofreading of your paper.
The linguistic revision was done by a medical editor and translator.
The main limit I see is the very small series, which you further divided according to the age group, leading to even smaller numbers in each group. Then you compared your results with a series of 762 patients. For most aspects, statistical significance was not obtained. I suggest you include some comments on these aspects.
This is an important consent. In the most aspect was no statistical significance. Nearly same results are shown in the researcher group from Taskinen.
The following 2 sentences were added to the manuscript:
The same results were seen by Jalkanen et al. in their study of adult patients with PUV who also considered their overall quality of life similar to that of the general population.
“…the correlation with a high number of subjects in the reference group of the KINDL questionnaire in particular only allows limited conclusions.”
Since your last patients was operated less than four years ago, I see this as medium term follow up.
I am not sure if we have understood the comment correctly. But the median follow-up was 64.5 months (IQR 36-97.25), the shortest follow-up is therefore correctly shown in the IQR. The correctness of the results was again confirmed by our Statistician.
Your conclusion is too generic, you should draw conclusions based on your study. Maybe answer at your question in the title of the paper.
We revised the conclusion.
Reviewer 3 Report
This is a retrospective study conducted on pediatric patients in a single institution between 2004 and 2019. The aim of the study was to evaluate the long-term outcome of ureterocystoplasty (UCP) in boys with posterior urethral valves (PUV).
Introduction is well written although I would write a few words regarding the impact of LUTS on QoL in general (10.3390/jcm11195639).
Matherials and Methods are clear and well described. Figure 2 is not visible and should be changed.
I have nothing to say regarding the results.
The discussion is extensive and include the most important points. Limitations are correctly reported.
Conclusions, although the sample is small, the conclusions are in line with the results. You should highlight the necessity of a larger sample and a prospective type of study in order to have more corroborating data.
Author Response
Dear Reviewer 3,
we highly appreciate the chance to further work on our manuscript in order to have it published in MDPI Children.
Your comments proved to be helpful and constructive, and we feel able to respond accordingly to each of them. Please find a point-by-point response, including information on the changes made to the manuscript below.
Comments are in italic, changes to the manuscript have been implemented by using the ´Track Change`function.
We would like to express our gratitude for your fair and thoughtful review, raising important points yet underestimated in our manuscript.
This is a retrospective study conducted on pediatric patients in a single institution between 2004 and 2019. The aim of the study was to evaluate the long-term outcome of ureterocystoplasty (UCP) in boys with posterior urethral valves (PUV).
Introduction is well written although I would write a few words regarding the impact of LUTS on QoL in general (10.3390/jcm11195639).
Thank you very much for this important suggestion. The impact of LUTS on QoL is a very important point in patients with PUV. The researcher group of Taskinen did some interesting research on this topic.
In our study we mainly focused on the general health related quality of life after ureterocystoplasty and therefore we didn´t use a specific questionnaire for LUTS. Therefore, we added the examination of general QoL in the introduction. Further on, we added the following sentences to the discussion:
“Health-related quality of life (HQoL) is still an underestimated topic in boys with PUV. This lack of consideration may be related to the fact that the reduction of the high mortality rate was only made possible by the improvements in prenatal medicine over the past years. Nowadays, boys with PUV are able to reach an age in which HQoL is of growing importance. In the past decade, the scientific group around Taskinen published some studies regarding HQoL, especially in adults with PUV [19,20]. The researchers found that the incidence of lower urinary tract symptoms (LUTS) is 2-fold higher in adult patients with PUV. Accordingly, general quality of life seems only to be impaired in case of urinary incontinence or renal insufficiency [19].
To achieve better objectivity, we decided to use the validated KINDL® questionnaire instead of a self-designed questionnaire. Yet, the evaluation of our patients is challenging because of the lack of specific questionnaires for boys with PUV or bladder augmentation. Unfortunately, the incidence of LUTS was not measured.”
Matherials and Methods are clear and well described. Figure 2 is not visible and should be changed.
Figure 2 was changed
I have nothing to say regarding the results.
The discussion is extensive and include the most important points. Limitations are correctly reported.
Conclusions, although the sample is small, the conclusions are in line with the results. You should highlight the necessity of a larger sample and a prospective type of study in order to have more corroborating data.
The following sentence was added at the end of the discussion:
“Nevertheless, a prospective study with larger sample sizes should be performed to reevaluate our results.”
Round 2
Reviewer 1 Report
The authors answered clearly and properly to my comments. Now the manuscript is suitable for publication.
Reviewer 2 Report
Thank you for making the suggested changes.